# Brief Communication: The Khurdopin glacier surge revisited – extreme flow velocities and formation of a dammed lake in 2017

Jakob F. Steiner[1], Philip D.A. Kraaijenbrink[1], Sergiu G. Jiduc[2], Walter W. Immerzeel[1]

[1]Utrecht University, Department of Physical Geography, PO Box 80115, Utrecht, The Netherlands
[2]Imperial College London, Centre for Environmental Policy, Faculty of Natural Sciences, SW7 1NA, London, United Kingdom

*Correspondence to*: Jakob F. Steiner (j.f.steiner@uu.nl)

**Abstract.** Glacier surges occur regularly in the Karakoram but the driving mechanisms, their frequency and its relation to a changing climate remain unclear. In this study, we use digital elevation models and Landsat imagery in combination with high-resolution imagery from the Planet satellite constellation to quantify surface elevation changes and flow velocities during a glacier surge of the Khurdopin Glacier in 2017. Results reveal that an accumulation of ice volume above a clearly defined steep section of the glacier tongue since the last surge in 1999 eventually led to a rapid surge in May 2017 peaking with velocities above 5000 m a$^{-1}$, which were among the fastest rates globally for a mountain glacier. Our data reveal that velocities on the lower tongue increase steadily during a four-year build-up phase prior to the actual surge only to then rapidly peak and decrease again within a few months, which confirms earlier observations with a higher frequency of available velocity data. The surge return period between the reported surges remains relatively constant at ca. 20 years. We show the potential of a combination of repeat Planet and ASTER imagery to (a) capture peak surge velocities that are easily missed by less frequent Landsat imagery, (b) observe surface changes that indicate potential drivers of a surge and (c) monitor hazards associated to a surge. At Khurdopin specifically, we observe that the surging glacier blocks the river in the valley and causes a lake to form, which may grow in subsequent years and could pose threats to downstream settlements and infrastructure in case of a sudden breach.

## 1 Introduction

Surging glaciers are not evenly distributed around the world's glaciated regions, but occur regularly under certain conditions (Sevestre and Benn, 2015). In the Karakoram, surges have been documented frequently since the end of the 19th century at numerous locations. Two general mechanisms driving surges were proposed: a build-up of ice volume during the quiescent phase in the reservoir zone of the glacier causing (a) increased basal shear stress resulting in till deformation at the glacier bed referred to as the *thermal switch hypothesis* (Clarke et al., 1984; Quincey et al., 2011), and (b) a collapse of hydraulic channels causing a switch from efficient surface and englacial drainage to sudden lubrication of the glacier bed referred to as the *hydrological switch hypothesis* (Kamb, 1987). Studies report surges in the region being controlled by both the first (Quincey et al., 2011) as well as the second mechanism (Mayer et al., 2011).

The Karakoram glaciers have received considerable scientific attention because of the anomalous regional mass balance (Kääb et al., 2015) and the large number of surging glaciers (Paul, 2015). Surging activity needs to be better understood to advance our knowledge of ice dynamic processes as well as glacially driven erosion and sediment transport in the region. Moreover, understanding of glacier surges is important as they may result in natural hazards that are due to the formation of ice dams and potential blockage of rivers.

Surges on Khurdopin Glacier, located in the Shimshal valley in Northern Pakistan (36°20'18"N, 75°28'3"E), have been documented to occur since the late 1800s and the most recent surges have occurred in 1979 and 1999 (Copland et al., 2011; Quincey et al., 2011; Quincey and Luckman, 2014; Rankl et al., 2014). These surges were characterized by a gradual

increase of velocities before the peak of the surge (Quincey and Luckman, 2014). During the surge events, the lower tongue is pushed further into the valley and has blocked the Vijerab River on several occasions, resulting in an ice dammed lake. In the region, a similar process has been observed and well documented for Kyagar Glacier (Round et al., 2017). Sudden drainage of the Khurdopin Lake has caused destruction to downstream villages before, which led to the development of an

early warning system with bonfires along the slopes of the entire Shimshal valley (Iturrizaga, 2005). So far these surges were solely described by investigating velocity data from distinct surface features of the glacier, using both coarse resolution satellite data, and field observations. Results show that the surge velocities can be up to two orders of magnitude faster than during the quiescent phase, however lack of cloud free imagery has made it difficult to accurately characterize the most recent surge (Quincey and Luckman, 2014). In this study, we put these earlier findings into further context by investigating a

new surge event in 2017 using recent satellite imagery. First we quantify surge velocities using automated feature tracking. We then quantify mass transport during quiescent and surge phases based on multi-temporal digital elevation model (DEM) analysis and we assess the potential hazard of lake formation using high-resolution optical satellite imagery. Finally, we discuss potential trigger mechanisms that may lead to the onset of the Khurdopin surge.

## 2 Data and Methods

To derive spatial velocities we use cross-correlation feature tracking using the COSI-Corr software (Leprince et al., 2007) on selected Landsat imagery between 2000 and 2017 (30 m resolution), and on Planet high-resolution imagery (3 m resolution) between 2016 and 2017 (Planet Team, 2017) (Supplemental Table S1). Volume changes were computed using the SRTM from 2000, a TanDEM-X DEM from 2011 and a DEM generated from ASTER imagery from May 2017. The ASTER DEM was generated using the open source Ames Stereo Pipeline software (Shean et al., 2016). We compared the DEMs in stable

off-glacier terrain and corrected the products accordingly (see Supplementary Material). Using the GlabTop2 model (Frey et al., 2014) and the SRTM, we computed ice thickness for the glacier and inferred the bed topography. Details on the specific COSI-Corr settings as well as the imagery used are provided in the Supplementary Material. The potential lake volume was calculated by intersecting the visually-derived lake perimeter with the TanDEM-X DEM.

## 3 Velocities during surge events

Khurdopin Glacier is approximately 41 km in length, 1.5 km in width and has an elevation range between 3300 m above sea level (a.s.l.) in the Shimshal valley to 7760 m a.s.l. It is heavily debris covered on the lower 10 km of the tongue and distinct meandering debris bands typical for surge type glaciers are present up to 20 km from the terminus. To investigate velocities on Khurdopin, we separated the tongue into 25 bins at 1 km equidistance along the centreline (Figure 1), and calculated the mean velocity within the bin. Using high-resolution imagery from the Planet satellites with sub-weekly overpasses (Planet

Team, 2017), we were able to characterize the surge event and the surface dynamics on the lower tongue and near the glacier terminus.

The surge of Khurdopin observed in 2017 confirms a recurring cycle typical for surging glaciers, ~20 years in this case, with observations of floods possibly caused by lake drainage after a surge in 1901 or 1904, 1923, 1944, 1960 (Hewitt and Liu, 2010) and observed surges in 1979, 1999 and 2017. Mean average surface velocities on the 25 km long main tongue of

Khurdopin during a quiescent phase are below 5 m a$^{-1}$, with a small peak of 15 m a$^{-1}$ at around 12 km along the tongue. The peak corresponds to a markedly steeper section of the profile (Figure 1). While lack of cloud-free imagery or poor image quality does not always allow accurate identification of the onset, peak and termination of the surge, the data suggest that a gradual increase of surface velocities over multiple years led to surge peaks with velocities up to 4000 m a$^{-1}$ in 1979 and 1999 (Quincey and Luckman, 2014). The most recent quiescent phase lasted from 2000 until at least 2011. By 2013 the

glacier had reached surface velocities above 100 m a$^{-1}$ beyond the steep section (km-12), but still smaller than 10 m a$^{-1}$ in the lower 5 km. The build-up phase between the quiescent phase and the actual surge peak between 2015 and 2016 was

characterised by increasing surface velocities in the tongue's upper reach (Figure 1 and Table S3 in the Supplementary Material). Between early 2017 and beginning of June velocities increased up to 5200 m a$^{-1}$ and dropped again to below 200 m a$^{-1}$ in most parts by September. While this extreme acceleration and deceleration happened within less than 9 months, the velocity peak along the longitudinal profile remained relatively stable (Figure 1). The gradual build-up and then relatively

short surge peak support earlier findings based on less frequently available data (Quincey and Luckman, 2014). While the available Landsat images were equally able to pick up the high velocities, it was only possible to characterize the actual surge development in such detail with the high-frequency Planet imagery, which additionally increases the chances for cloud-free imagery.

## 4 Ice volume changes during surge events

Apart from increased velocities, surges logically also result in large amounts of displaced ice volume. In many cases this results in a rapid extension of the positon of the glacier's snout. However, in the case of Khurdopin the apparent terminus does not advance and has not done so during at least the recent surges, since it has turned into a stable moraine, dynamically decoupled from the active part of the glacier. This makes detection of actual length changes of the active tongue visually difficult (Figure 1). Using three DEMs (SRTM in 2000, TanDEM-X in 2011 and ASTER in 2017; Supplementary Table S2)

the elevation change rates for the quiescent and surge phases are quantified (Figure 2). To match the ice volume changes with the actual surge phase, we extrapolated the annual rate of surface elevation change observed between 2000 and 2011 until the 28$^{th}$ of August 2015 as a proxy for the quiescent phase surface elevation change. Given the steep rise in velocity it is assumed that from the 28$^{th}$ of August 2015 onwards the surge phase started and the annual rate in surface elevation change was estimated from the remaining volume difference and the 2017 DEM. The transition from positive to negative elevation

change during the quiescent phase is clearly notable and coincides with the steep section of bedrock around km-12 (Figure 2, panel a), an observation made earlier by Gardelle et al., (2012), who identified this distinct behaviour for other glaciers in the region as well. This distinction is again visible exactly at the same location for the surge, when elevation change is positive in the lower reach where mass is accumulating. During the surge in May 2017 the glacier surface between km-3 and 12 has gained height by 50 to 160 m. Based on the elevation changes we find a net volume gain between 2015 and 2017 of 2485 ·

10$^6$ m$^3$ (+/- 55 · 10$^6$ m$^3$ based on the DEM accuracy) between the steep section and the part of the terminus where no more surface change is visible. Averaged over the entire glacier we estimate that the overall volume loss is slightly negative (see surface elevation change in Figure 2), similar to what is reported by Bolch et al., (2017).

## 5 Hydrology and Hazards

The tongue of the Khurdopin Glacier reaches across the main valley floor. As a consequence the glacier has blocked the

local Vijerab River multiple times in the last century, caused by the tongue pushing towards the opposite headwall of the main valley (Figure 3). Most of the reported lake drainages were not catastrophic and they have rarely caused damages downstream beyond eroded fields and damaged bridges (Hewitt and Liu, 2010; Iturrizaga, 2005). From historic Landsat imagery it is obvious that a lake formed during the melt season in two consecutive years after the surge in 1999, likely because the added mass required considerable time to be eroded. In late April 2017, the lake formed at exactly the same

location, growing quickly from 72000 m$^3$ at the beginning of May to 1 · 10$^6$ m$^3$ one month later and peaking at 2 · 10$^6$ m$^3$ on the 28$^{th}$ of June. The lake finally drained starting around the 21$^{st}$ of July and had disappeared by 5$^{th}$ of August. As a consequence the river washed away the road at multiple locations, destroyed at least one main bridge and eroded local agricultural land, making the valley inaccessible for a week. Ice floes on the water surface indicate ice calving from the advancing tongue and could pose an additional threat as they could block a drainage channel temporarily and create a sudden

spill upon disintegration. Considering the height of the advanced glacier tongue – between 15 m at the fringe and up to 160 m on the surging tongue – and the fact that in 2000 the lake reached lake levels ca. 10 m higher than in 2017, we show

potential lake extents that could reach beyond 1 $km^2$ or $10 \cdot 10^6\,m^3$, possibly during the melt season of 2018 or 2019. Repeat floods in the one or two years after a possible surge event have been reported multiple times in the recent century as well (Hewitt and Liu, 2010). The volumes calculated could be decreased by sediments visibly deposited either by the surging glacier or the dammed Vijerab River.

## 6 Discussion and Conclusion

The data collected and analysed support earlier studies on Khurdopin in the observation of a relatively constant return period of a glacier surge of 20 years since the end of the 19[th] century, irrespective of a changing climate and surges of nearby glaciers (Hewitt and Liu, 2010; Quincey et al., 2011; Quincey and Luckman, 2014). Using distributed velocity and elevation change data we furthermore show that a division point exists at 12 km up-glacier that separates two distinct reaches of the tongue: (a) the upper reach where velocities gradually increase during the build-up phase and mass continuously accumulates during the 19 years between surges, and (b) the lower reach where velocities peak during the surge and the ice mass previously accumulated in the upper reach is relocated within only a number of weeks. This line likely coincides with a steep bedrock section and is located just below a tributary that possibly supplies a lot of additional mass via avalanche deposits. The surge of 2017 showed a similar four-year build-up time as the surge in 1979 over which the glacier surface in the upper reach increased by approximately 3 m $a^{-1}$ and decreased by up to 7 m $a^{-1}$ in the lower reach. This period is defined by constantly increasing velocities in the upper reaches. It is difficult to ascertain which are the main drivers for the regular surges on Khurdopin Glacier (Quincey and Luckman, 2014). In combination with a gradual accumulation of mass on the upper tongue during quiescence and a resulting steepening surface gradient the actual surge starts rapidly when a tipping point is reached. Ice deformation $u_d$ (Greve and Blatter, 2009; Round et al., 2017), with a modelled ice thickness between 120 m and 350 m, results in a velocity of 1 to 60 m $a^{-1}$ for the 1 - 4° steep surface gradient over the whole tongue and 3 m $a^{-1}$ at the steep section. This is in the same order of magnitude as the measured velocities during quiescence and the early build up phase. While these values stay relatively stable for most of the tongue and can account for the overall glacier flow velocities they increase by an order of magnitude to more than 50 m $a^{-1}$ in the steep section due to the increase in surface gradient, and thus making up more than 50% of the observed surface velocity. The increase in velocity at this specific location has possibly caused a switch from an otherwise cold to a temperate bed, initiating a rapid increase in basal sliding from 2015 onwards. The sudden absence of supraglacial ponds on the terminus during the surge (Figure 3) and the formation of a supraglacial pond in May 2000 after the last surge exactly at the location of the clear line of change around km-12, could also point at a disturbed englacial network playing a role (Kamb, 1987; Mayer et al., 2011). At least the last two surges occurred at the beginning of the melt season, which could further catalyse the surge if melt water reaches the ice-bedrock interface. Basal sliding is also most likely the dominant flow process as the cross profiles of surface velocity indicate plug flow, characterized by flat rather than parabolic velocity profiles as was observed during the quiescent phase (Kamb et al., 1985). As previously suggested the surge on Khurdopin is hence likely triggered by the thermal switch but the actual surge is dominated by basal sliding (Quincey and Luckman, 2014), similar to Kyagar Glacier (Round et al., 2017). Future field observations should focus on finding possible evidence for these processes and possible feedback processes, especially related to the deformation of water-saturated granular base material that could explain these extreme acceleration rates and peak velocities (Damsgaard et al., 2015). The surface velocities observed during the peak surge in May 2017 on Khurdopin Glacier are, together with the recently observed surge on the neighbouring Hispar Glacier (Paul et al., 2017), the fastest so far reported for the region. In their magnitude and rapid acceleration and deceleration they are comparable to similar bursts at the closely investigated Variegated Glacier (Kamb et al., 1985), where observations with even higher temporal resolution were available. The increased velocity and associated ice volume redistribution resulted in increased strain rates, evidenced by crevasses appearing at the glacier surface since early May with a marked increase in size and number since mid-June (Figure 3d). The high peak rates of basal slip can result in erosion rates up to 0.5 m $a^{-1}$ for a brief period (Humphrey and

Raymond, 1994), a value several order of magnitudes higher than typical erosion rates in mountain ranges. Large amounts of additional sediments were visible at the glacier snout during the surge.

We can show that newly available satellite imagery with multiple cloud-free sub-weekly image pairs makes the characterization of such a rapid surge cycle possible and confirms high peak velocities that are easily missed by less frequently available Landsat imagery. As a consequence of the surge a lake has formed in the proglacial valley, similar to earlier surges. We quantified its evolution and potential future expansion as it is very likely that the lake will reappear during melt seasons in the following two years until the accumulated mass has sufficiently eroded for the water to drain freely. Exploiting the potential of only recently available high-resolution imagery with frequent overpasses could lead to a better understanding of such surges as it provides the potential for more accurate velocity data (Altena and Kaab, 2017). Additionally, it would also enable faster assessment of risk potentials and subsequent warning of affected communities.

## 7 Author Contributions

JFS, PDAK and WWI designed the study, JFS and PDAK carried out the data analysis, JFS wrote the manuscript. SGJ pointed out the occurrence of the surge and provided contacts to the local authorities. PDAK, SGJ and WWI reviewed the manuscript.

## 8 Acknowledgements

This project was supported by funding from the European Research Council (ERC) under the European Union's Horizon 2020 research and innovation program (grant agreement no. 676819) and by the research programme VIDI with project number 016.161.308 financed by the Netherlands Organisation for Scientific Research (NWO). We would like to thank Mr. Waheed Anwar for pointing out the start of the surge in April and for providing the photos included in the manuscript. We also would like to thank PlanetLabs for providing access to their high-resolution imagery.

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

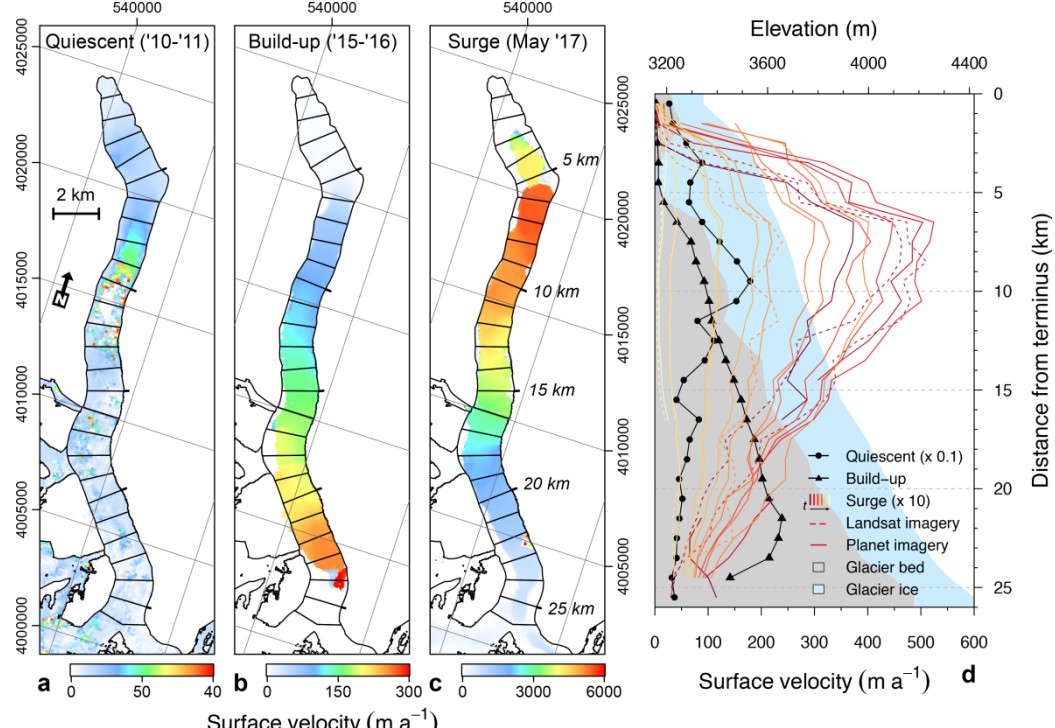

Figure 1: Velocities measured from cross-correlating Landsat imagery of one year of the quiescent phase (a; 17th of October 2010 – 18th of September 2011), the last year of the build-up (b; 28th of August 2015 – 10th of May 2016) and the surge peak in May 2017 (c; 13th to 29th of May, 2017). Panel (d) shows mean values of the bins compared against bed elevation and all available velocity pairs for 2017, between December 2016 (dark red) and September (yellow), both for Landsat and Planet. Values for quiescence are shown at one order of magnitude larger, values for the surge phase at one order of magnitude smaller than measured. Note the difference in scales for the different phases. An animation of the images used to derive the velocities can be found in the Supplementary Material. All coordinates are in UTM WGS84 Zone 43.

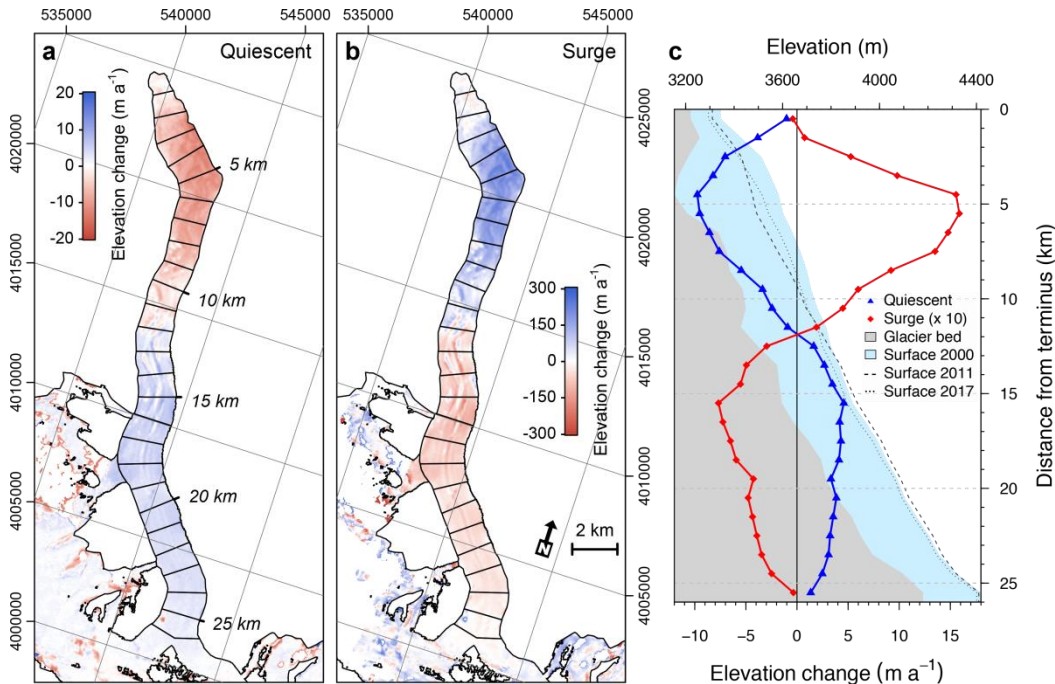

Figure 2: Elevation change rates during the quiescent phase (a), and during the build-up and surge phase (b). Mean values per bin are shown in panel (c). Note that values for the surge phase are shown at one order of magnitude smaller than measured. All coordinates are in UTM WGS84 Zone 43.

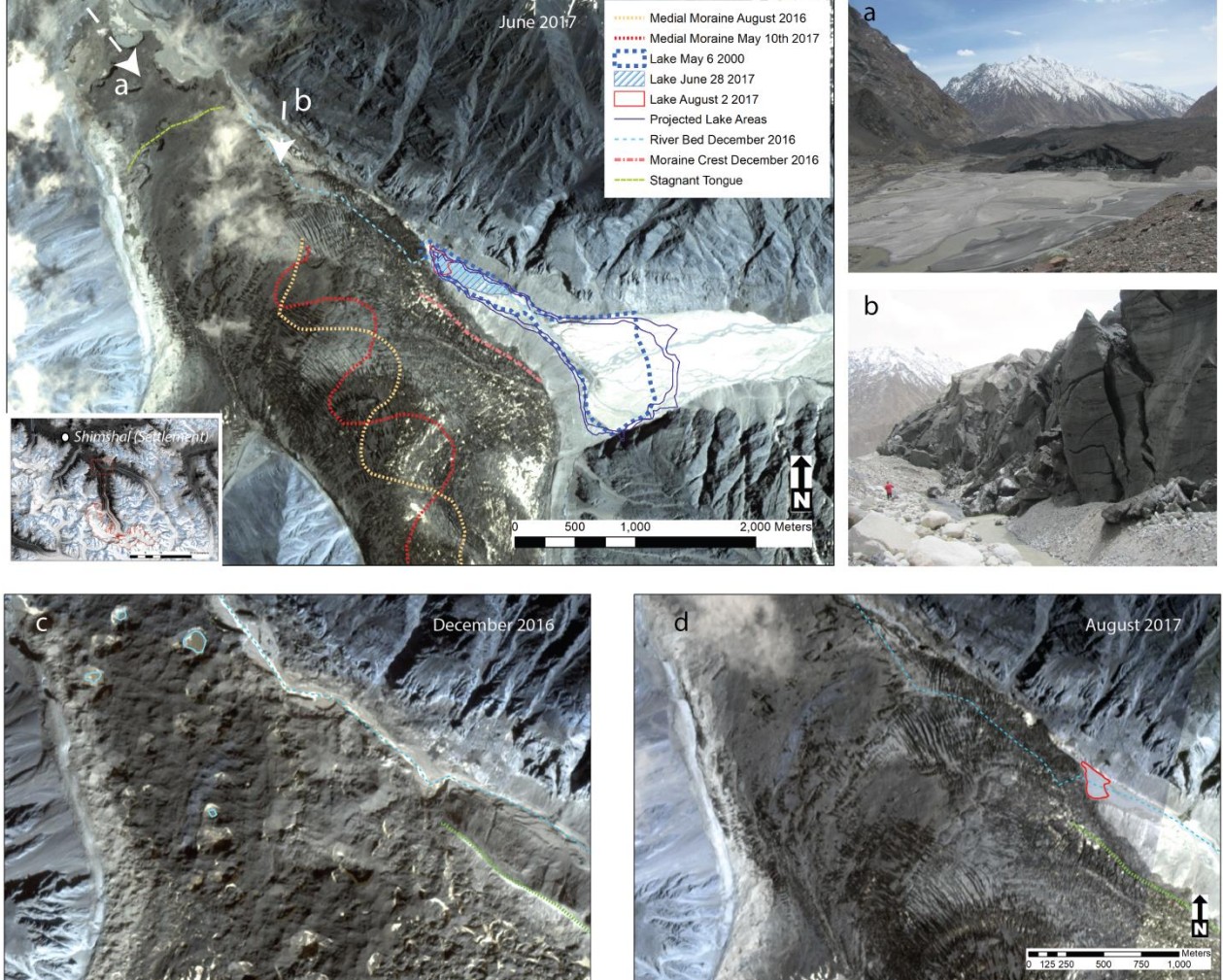

**Figure 3: Evidence from the surge event visible at the tongue. The main panel is based on a Planet image from the 28th of June 2017** (Planet Team, 2017)**. The medial moraines, the lake extent in 2017, the original river bed and moraine crest are also mapped from Planet imagery. The lake extent in 2000 is mapped from the panchromatic band of Landsat 7. The projected lake extents and depths are computed based on the TanDEM-X. Arrows at (a) and (b) denote angle of view for images on the right. Panel (a) shows an overview of the front of the tongue and panel (b) shows the front of the advance. Note the fine dark sediments often associated with a surge event. The tongue below the dashed green line showed no change during the surge. Panels (c) and (d) show the tongue surface before and after the surge respectively. (Photos: Waheed Anwar, May 2017)**