# Peer review of "Brief Communication: The Khurdopin glacier surge revisited – extreme flow velocities and formation of a dammed lake in 2017"

_The Cryosphere, 2017_

## Referee Comment (RC1) · V. Round (Referee) · 21 Aug 2017

This brief communication describes the recent surge of the Khurdopin glacier through remotely-sensed observations of surface velocity and surface elevation, and highlights the extreme flow velocities and development of a potentially hazardous ice-dammed lake. The new velocity observations are consistent with previous surges of Khurdopin Glacier, for instance the 18 year period since the last surge, the pattern of acceleration before the surge and the season and magnitude of the peak surge velocities.

General comments:

[Figure]

The manuscript is well written and the methods sound, but aside from confirming similar behaviour to previous surges and alerting us to possible lake formation, more could be done to communicate what is new or novel about the results of this study. The following are the main points which I think have potential to be expanded upon to increase the value of the study.

1) The Planet satellite data used here has much higher temporal and spatial resolution than the Landsat data which has been used in previous studies. Does this provide any particular benefit for the analysis of the velocities over the course of the recent surge?

2) The DEM analysis nicely shows the expected patterns of mass displacement during quiescence and surging. This is new for specifically surge related studies of this glacier, but similar results have already been seen for Khurdopin glacier as part of mass balance studies by Gardelle et al. (2012) and Bolch et al. (2017). Do the results presented here provide any new insight into why or how the glacier surges?

3) The relationship between the surge and recent lake formation is explained and some projections of potential lake area and depth are made. I think that these projections could be communicated with a little more detail, for example what factors may affect the likelihood of the lake reaching the projected sizes or why this lake could be particularly relevant. Is the hazard expected to be greater than after the previous surges, for which there were no recorded damages according to Iturrizaga (2005)?

4) The hypothesis that the 'thermal switch' mechanism is responsible for the surge initiation doesn't seem to be backed significantly by the results, or if so the discussion of this assumption needs to be strengthened. If there isn't significant backing for either of the often cited 'thermal switch' or 'hydrological switch' initiation mechanisms, then I don't think it is necessary to classify the surge as either.

5) The 'Velocities during surge events' section gives a long description of the velocity changes in time and space, but it is difficult for the reader to build an overview picture. A visual summary of the temporal evolution would help greatly, especially given the

large amount of data available. I think this is important if you want to talk about the 'advance of the surge front' (P3,L3). Figure 1 shows the three extremes of the surge but not the evolution between.

Specific comments:

P1, L12: 'during a glacier surge in of' remove 'in'

P1, L14: Does the 'fastest rates globally' refers to peak rates for surging glaciers or to glaciers in general? The use of 'm/a' in '5000m/a' could suggest that rate to be an average rate over a year. Using "m/d" may avoid this confusion.

P1, L15-16: This sentence could do with some reworking. Firstly it isn't clear if the four year build up in velocity occurred over the whole glacier or just part of it. Also the term 'upper tongue' probably has little meaning to the reader at this stage.

P1, L19: The 'however' at the beginning of the sentence implies contradiction of the hypothesis in the sentence before. Does the crevassing and disappearance of supra glacial ponds contradict the thermal switch mechanism, or reduce certainty in your hypothesis? I think these observations indicate a factor which amplifies the surge regardless of the initiation mechanism.

P1, L27: I suggest a slight rewording of the two general driving mechanisms, because the 'build-up of ice mass during the quiescence phase. . .' applies to both mechanisms.

P2, Section 2: There could be a few more details in the methods section here instead of only in the supplement. I would like to see at least an indication of the temporal resolution/number of images from Landsat and Planet (this is also missing from the supplement). Perhaps also spatial resolution (what is meant by 'high resolution') and/or indication of error margins.

P2, L16: SRTM should be mentioned here too as it was also used for investigating mass changes.

P2, L23: Is this information about the source of the debris/medial moraine included because it is important to the glacier velocity?

P2-3, Section 3: I tended to get lost reading this section with its rather long chronological description of the three surges. One could present the results by describing the various phases of the three surges simultaneously. This could cut out some repetition and make similarities more apparent. Displaying this information about the temporal evolution of velocity as a figure would also allow the text here to be shortened and provide a very valuable summary and overview of the surges. Velocity over time could be shown for both the lower and upper parts of the tongue, as these show different behaviour, or better still for the whole length of the tongue.

P2-3, Section 3: Is the difference in peak velocities between the different surges, e.g 2000m/a in 1999 and 5000m/a in 2017, a real result or could it be an artefact of the temporal averaging period, where shorter periods are more likely to capture faster peak velocities?

P3, L3: The advance of the 'surge front' is not clear to me. Quincey et al. (2011) show a very distinctive surge front at Kunyang Glacier but not for the 1999 surge of Khurdopin Glacier. Citing the surge front observed by Quincey et al. (2011) implies a similar acceleration pattern to the Kunyang surge. Perhaps the term 'surge front' is a bit subjective in this case. This is where a visual representation of the temporal changes, with more than three times steps, would be really useful.

P3, L9: The comment about not being able to discern length change is repeated in Section 4. I would expand upon it here and remove from section 4, or just remove it here.

P3, Section 4: DEM differences between 2000 and 2008 were calculated for Khurdopin glacier also by Gardelle et al. (2012), I think this paper is definitely worth consulting as they also focus on Khurdopin glacier for getting ablation rates. (Gardelle et al. 2012, Slight mass gain of Karakoram glaciers in the early twenty-first century, Nature

Geoscience Letters, DOI: 10.1038/NGEO1450).

P3, L22: Was the mass change over the whole glacier assessed between 2000 and 2011, or just 2011 and 2016? Is there enough confidence in the results to give us a number for these periods?

P3, L26-27: This sentence makes is seem like there have been considerable damages in recent decades, but Iturrizaga (2005) shows most damages in the early 1900s. Is there another source showing more recent damages, or is it possible that the floods have become less severe or the settlements less vulnerable?

P3, L29: The lake outbursts at Kyagar glacier discussed by Haemmig et al. (2014) were extremely rapid, jökulhlaup type events, not gradual as mentioned here.

P3, Section 5: The potential lake volumes might have more meaning for hazard assessment than the surface area. I imagine this could be quite easily calculated given the DEM of the lake basin.

P3, L36: I assume the 15 meter height increase at the fringe represents the upper bound on potential lake depth. Is there any indication that this height will increase or decrease in the next couple of years and what factors might affect the likelihood of the lake reaching these various levels? Additionally, the 80 meter increase at the centre doesn't seem relevant for the lake.

P3, L37: Do you mean the potentially large influx of subglacial sediments is into the potential lake basin? What effect would this have on the lake - decrease the potential volume of the lake?

P4, L2: Two surge periods is probably not sufficient to confirm 'a constant return period' especially over longer timescales, unless there are earlier indications of similar return period.

P4, L5: The observation of very different behaviour of the lower and upper parts of the tongue, separated by a steep part of bedrock at 12km, is interesting and has been

observed on other surging glaciers (Quincey et al. 2014, Round et al. 2017). Possible questions to discuss here are whether there is something about the lower part of the tongue that leads it to experience such extreme changes in behaviour, or what the significance of the steep section at 12km may be, or the significance of the avalanche mass deposits?

P4, L12-13: Couldn't the increased pressure and 'tipping point' reached at the end of the quiescence also initiate the surge through collapse of the subglacial drainage system or failure of subglacial till? Is it possible to distinguish between these processes with the available data, or is there some other indication leading to the conclusion of a switch from cold to temperate basal conditions?

P4, L12-13: Do you mean this switch from cold to temperate based applies to the upper part of the tongue with the gradual acceleration, or lower part of the tongue with the sudden surge acceleration? Is it feasible that the velocities during the assumed cold based phase be purely due to ice deformation?

P4, L14: Quincey and Luckman (2014) suggested both the 'thermal switch' or 'subglacial drainage' as possible controls and didn't seem to have enough evidence to conclude one way or the other.

P4, L20: Did the velocity results show a parabolic velocity profile across the tongue during the quiescence? This wasn't mentioned in section 3 but would be interesting..

P4, L21: The peak velocities of this surge are really incredibly high, as is the magnitude of the acceleration! A mention some of the feedback processes which could allow such extreme basal sliding velocities could be informative. Do you think subglacial till deformation plays much of a role?

P4, L26: I'm not sure how the increased resolution and overpass frequency of the Planet satellite data have led to better understanding of the surge. Is it the ability to resolve the peak velocity over shorter time frame or observation of more temporal

fluctuations or spatial patterns (e.g. transverse variation) in velocity? If so then this should be discussed somewhere.

Figures 1 and 2: The right hand panels show the inferred glacier bed elevation, however it would make sense to also show the observed glacier surface elevation. Showing the surface elevation from the 2011 and 2017 DEMs would provide an additional visualisation of the mass redistribution, and if shading or dashed lines are used the readability of the plot shouldn't be affected.

Figures 1 and 2: The maps should be in some way georeferenced.

Figure 3: Very nice to have some photos from the ground, but maybe indicate the date (month)

Figure 3, L3: The traced 'centreline' would more appropriately be referred to as 'former centreline' or 'former medial moraine'

Figure 3, L8: I would say the tongue below the dashed green line "showed no change during the surge" rather than "remained stable".

Supplement Table S1: How many images were used from each Satellite? It would be interesting to have this information about the potential temporal resolution of the data.

Supplement Table S1 (DEM data): This table should be labelled Table S2.

Supplement Table S2: The SRTM from 2000 should also be shown here as it was also used for the surface elevation analysis.

Supplement: The COSI-Corr model setup and potential error magnitude is explained clearly.

---

## Referee Comment (RC2) · D.J. Quincey (Referee) · 21 Sep 2017

This paper presents surface elevation change and surface velocity data from before and during the recent surge of Khurdopin Glacier in the Shimshal Valley of Pakistan. These data are used to characterise the surge evolution, calculate mass change, quantify the surge return period, and describe the evolution of an ice-marginal lake. The manuscript is generally well-written and the data contained within are new and interesting. There are a few areas where with only a small amount of further work the manuscript could be improved – these are detailed immediately below – followed by some more minor comments that should provide some more clarity in places.

1. The key take-home message is currently a bit hidden. It seems to me that the new findings are: 1. that the surge return period appears to be of the order of 20 years (whilst acknowledging that n=2); 2. that surge velocities may be even faster than previously realised – implications for erosion and sediment transport; 3. that there may be a topographic control on this particular surge (but this needs much greater discussion – see following point); 4. that the ice-marginal lake is posing a hazard to local communities. If the abstract and the conclusions could be modified to give the key message much greater prominence the manuscript would have greater impact.

2. The relevance of the steep bed topography at 12-km needs some further discussion/explanation. Is the suggestion that it provides a control on surge dynamics? Or even that it is responsible for the spatial imbalance in flow? Presumably it doesn't provide a bottle-neck to flow (I imagine the opposite if anything)? Is the modelled ice particularly thin above the step and potentially frozen? Some consideration of the possibilities would be a welcome addition.

3. There appear to be many more velocity datasets discussed in the text than presented in the figures. Is there a reason for not showing all of the velocity data? It would really help with visualising the evolution of the surge to have them all (or at least more than the current three) available.

4. The discussion of whether the surge is thermally or hydrologically triggered lacks real evidence so I would suggest toning it down or even removing it. It is likely that both thermal and hydrological processes will be at play as you infer in your own discussion.

5. There needs to be some uncertainty analysis of the dh/dt data. How well co-registered were the DEMs? Showing off-glacier areas of dh/dt data (and velocity data) would help here, as would the distribution of those values. This extra analysis would be a good addition to the Supplementary, with uncertainty shading added to the figures and an error range added to the values stated in the main text.

Minor comments:
P1 12: 'during a surge of the Khurdopin Glacier in 2017.' (also elsewhere, glacier should be Glacier where you are referring to it by name).

P1 15-16: I'm not sure there is evidence for a surge front in the data you show here?

P1 19-20: do you show these surface observations? It's difficult for the reader to believe the extra lubrication suggestion without seeing evidence.

P1 26: this is maybe misleading. . . has an increase in frequency been reported? Or just an increase in number? And is that not because we have better and better data? Without repeat datasets (like those presented here) we can't say for sure whether frequency is increasing or not.

P1. 34-35: what do you mean by 'understanding regional glacier behaviour'? Is 'in order to advance knowledge of basal processes, non-steady flow more generally, and erosion and sediment transport in the region' perhaps a better justification?

P2 3: name the glacier here, and also specify in the next sentence that it's the Khurdopin lake (not Kyagar) that has previously caused destruction.

P2 8: maybe 'recent' is better than 'novel' here? Novel implies something a bit different about it.

P2 9: do you actually quantify the mass transfer somewhere? I don't see it. . .

P2 16: was the ASTER DEM derived by USGS? Or by the authors? In either case, some further information is required about its expected vertical accuracy and how well it performs against the TDX DEM.

P2 22: can you add the value (of mass loss) here?

P2 23: is it subglacially sourced for sure? I've always imagined it to be plucked from the spur where the two main tributaries meet.

P2 26-27: is there a reason why you don't show these finer resolution velocity data?

P2 31: maybe reword to 'does not always allow the onset, peak and termination of the surge to be accurately identified, the data suggest that...'?

P2 32: not sure 'build-up' needs italics (here or at line 38)?

P2 32-40 change to past tense here ('were below... and quickly rose... increased in 1998... and peaked in spring 1999... phase lasted until... glacier had reached... was characterised by... velocities had reached... had further accelerated')

P3 7: I'm not sure Figure 3 really supports this statement...

P3 9 and 14: if the lowermost 1 km of the glacier is not impacted by the surge is the length change not zero? What is meant by length change here (if not position of the terminus)?

P3 20-21: this is a long section between the commas – consider moving 'at rates comparable to those of the quiescent phase' before the first comma

P3 32-38: it should be a short step to calculate the volumes from the DEM data – these values would be a valuable inclusion here.

P4 21: not quite true. The recent Hispar paper (doi:10.3390/rs9090888) by Paul et al. show comparable velocities

P4 25-27: as far as I can tell the Planet imagery did not contribute to the data you present here other than the overview in Figure 3.

Figure 1: some co-ordinates either here or in the text would help readers not familiar with the glacier to locate it.

Figure 3: I'm not sure the wiggles are best described as 'centrelines'? Are they not the contorted medial moraines that have shifted position?

Supplementary: can you provide the image tile names in each case?

Supplementary: Table S1 should be S2 in second case (and should SRTM be included

here?).

Supplementary: the animation is excellent. Should it not be referred to somewhere in the text (or it may go largely un-noticed. . .)?
* * *

---

## Author Comment (AC1) · 11 Oct 2017

Response to Reviewer

Manuscript: Brief Communication: The Khurdopin glacier surge revisited – extreme flow velocities and formation of a dammed lake in 2017

Reviewer: V. Round
* * *
We greatly appreciate the concerns raised by the Reviewer and respond to each of them below. We have extended the discussion related to the processes and possible explanations and now provide greater detail in the discussion. We agree that we could expand even more on some details, but wanted to keep this study focused on this particular event and within the "Brief Communication" format.

Original comments by the reviewer are in bold, followed by our response. Note that the page and line number are always given twice, once for the document with markups which is provided at the end of the Response, and once to the revised manuscript without markups, which will be provided later.

**P1, L12: 'during a glacier surge in of' remove 'in'**

Thanks for pointing this out, it is now amended.

**P1, L14: Does the 'fastest rates globally' refers to peak rates for surging glaciers or to glaciers in general? The use of 'm/a' in '5000m/a' could suggest that rate to be an average rate over a year. Using "m/d" may avoid this confusion.**

This does indeed refer to surging glaciers, as there are other examples with similar or higher velocities such as the Lowell Glacier (Bevington and Copland, 2014) and the Variegated Glacier (Kamb *et al.*, 1985), however these extreme values (65 m d$^{-1}$) were measured just during a 2h rather than a daily interval.

We think that switching between units could be confusing and makes these values less easily comparable to other reported values in the region which are generally given in m a$^{-1}$. By referring to the month of May specifically it should be clear that this only refers to this period. However we refer to the importance of frequent image availability to derive such high velocities now on P6L7 / P4L39ff.

**P1, L15-16: This sentence could do with some reworking. Firstly it isn't clear if the four year build up in velocity occurred over the whole glacier or just part of it. Also the term 'upper tongue' probably has little meaning to the reader at this stage.**

We have reworded this part to increase clarity.

**P1, L19: The 'however' at the beginning of the sentence implies contradiction of the hypothesis in the sentence before. Does the crevassing and disappearance of supra glacial ponds contradict the thermal switch mechanism, or reduce certainty in your hypothesis? I think these observations indicate a factor which amplifies the surge regardless of the initiation mechanism.**

Thanks for pointing this out. We have amended this accordingly.

**P1, L27: I suggest a slight rewording of the two general driving mechanisms, because the 'build-up of ice mass during the quiescence phase' applies to both mechanisms.**

Thanks for pointing this out. We have adapted this accordingly to make it applicable to both.

**P2, Section 2: There could be a few more details in the methods section here instead of only in the supplement. I would like to see at least an indication of the temporal resolution/number of images from Landsat and Planet (this is also missing from the supplement). Perhaps also spatial resolution (what is meant by 'high resolution') and/or indication of error margins.**

We have now provided all details for the images used in the Supplementary as well as error ranges for the DEMs and how they were derived. Since the space for a Brief Communication paper is limited we do not want to go to great lengths in the methods section, as the approaches used are quite straight forward and well documented in the literature.

**P2, L16: SRTM should be mentioned here too as it was also used for investigating mass changes.**

The reviewer is right, this was an omission and we have added it in P2L26 / P2L19 of the revised manuscript.

**P2, L23: Is this information about the source of the debris/medial moraine included because it is important to the glacier velocity?**

The reviewer is right, this information – although interesting from a glacier flow point of view and the erosion potential of surge type glaciers – is not so relevant to this specific study. We have therefore omitted it in the revised manuscript.

**P2-3, Section 3: I tended to get lost reading this section with its rather long chronological description of the three surges. One could present the results by describing the various phases of the three surges simultaneously. This could cut out some repetition and make similarities more apparent. Displaying this information about the temporal evolution of velocity as a figure would also allow the text here to be shortened and provide a very valuable summary and overview of the surges. Velocity over time could be shown for both the lower and upper parts of the tongue, as these show different behaviour, or better still for the whole length of the tongue.**

We agree that this is a bit convoluted and have decided to specifically focus on the latest surge and leave the analysis of earlier surges to the already published work. We only refer to the similarities in behavior and we have included all velocity profiles of the surge in figure 1 and all velocity data in table S1 in the supplementary material, which should visualize the surge development in a more concise way.

**P2-3, Section 3: Is the difference in peak velocities between the different surges, e.g 2000m/a in 1999 and 5000m/a in 2017, a real result or could it be an artefact of the temporal averaging period, where shorter periods are more likely to capture faster peak velocities?**

Indeed it is quite likely that it's the more frequent availability of satellite images that makes it possible to only see these high peak velocities now. As the reviewer suggested above, this potential of Planet imagery should be further emphasized in the manuscript and we have discussed this in the Discussion at P6L7 / P4L39ff.

**P3, L3: The advance of the 'surge front' is not clear to me. Quincey et al. (2011) show a very distinctive surge front at Kunyang Glacier but not for the 1999 surge of Khurdopin Glacier. Citing the surge front observed by Quincey et al. (2011) implies a similar acceleration pattern to the Kunyang surge. Perhaps the term 'surge front' is a bit subjective in this case. This is where a visual representation of the temporal changes, with more than three times steps, would be really useful.**

Thanks for pointing this out. We agree that "front" may have been used too subjectively and we have rephrased it. As the reviewer suggested we have now added all velocity data from the surge to figure 1 which shows the development of the surge and also that the front advances slightly but that the peak of velocity actually remains nearly in the same position. The data therefore support earlier observations of Khurdopin without a clearly identifiable surge front (Quincey and Luckman, 2014).

**P3, L9: The comment about not being able to discern length change is repeated in Section 4. I would expand upon it here and remove from section 4, or just remove it here.**

Thank you for pointing this out, this is indeed redundant. As it fits better in section 4, we have removed it here.

**P3, Section 4: DEM differences between 2000 and 2008 were calculated for Khurdopin glacier also by Gardelle et al. (2012), I think this paper is definitely worth consulting as they also focus on Khurdopin glacier for getting ablation rates. (Gardelle et al. 2012, Slight mass gain of Karakoram glaciers in the early twenty-first century, Nature Geoscience Letters, DOI: 10.1038/NGEO1450).**

Thanks for pointing this out. We agree that this is important in this context and have added it in P4L10 / P3L22.

**P3, L22: Was the mass change over the whole glacier assessed between 2000 and 2011, or just 2011 and 2016? Is there enough confidence in the results to give us a number for these periods?**

The DEM differencing is for the period between 2011 and 2017, hence the total change in elevation is simply inferred. We have therefore adapted this and defined the possible range from dH = 50 m if we assume the tongue to have had a net mass change of zero in the build up phase to dH = 80 m if we assume that the net mass loss was equal to the quiescent phase.

**P3, L26-27: This sentence makes is seem like there have been considerable damages in recent decades, but Iturrizaga (2005) shows most damages in the early 1900s. Is there another source showing more recent damages, or is it possible that the floods have become less severe or the settlements less vulnerable?**

The reviewer is right that the more serious damages reported by (Iturrizaga, 2005; Hewitt and Liu, 2010) were reported before the 1960s. We have therefore removed this part and mentioned the level of damage in P4L35 / P3L39. It is difficult to say whether damages in recent decades have increased or decreased. While people may be less vulnerable today or better adapted to possible floods, infrastructure has also increased and people are more used to the fact that there is a road to the main Hunza valley. Indeed during the flood that occurred in 2017, one main bridge was destroyed and the road connecting the valley to the outside world blocked for a week (http://pamirtimes.net/2017/08/01/shimshal-river-flood-bridge-destroyed-road-damaged-cultivable-land-affected-at-several-places/).   Unfortunately

both discharge stations installed were destroyed during the flood, making peak flood measurements impossible.

**P3, L29: The lake outbursts at Kyagar glacier discussed by Haemmig et al. (2014) were extremely rapid, jökulhlaup type events, not gradual as mentioned here.**

Thanks and we have adapted the text accordingly in P4L37ff / P3L31ff. We have also added some comments on lake drainage, resulting from the actual drainage of the lake.

**P3, Section 5: The potential lake volumes might have more meaning for hazard assessment than the surface area. I imagine this could be quite easily calculated given the DEM of the lake basin.**

Thank you for the suggestion, we have now made volume estimates and discussed this in the paper.

To explain how the volume was derived we added a line in the Methods on P2L27 / P5L5 and we describe this in the supplementary material in section 3.

**P3, L36: I assume the 15 meter height increase at the fringe represents the upper bound on potential lake depth. Is there any indication that this height will increase or decrease in the next couple of years and what factors might affect the likelihood of the lake reaching these various levels? Additionally, the 80 meter increase at the centre doesn't seem relevant for the lake.**

We have improved the explanation and adapted the Text as well as the projected lake areas. 15 m are the approximate cliff height which presumably can act as the dam for potential future lakes. While water may pond beyond that on the tongue itself whether and how this could happen we do not know and we have therefore removed it. We now just use the former lake from 2000 as well as the ice wall as an indicator of possible future extents.

**P3, L37: Do you mean the potentially large influx of subglacial sediments is into the potential lake basin? What effect would this have on the lake - decrease the potential volume of the lake?**

The possible sources include the subglacial erosion of Khurdopin but also the sediment carried in from the Vijerab river. This would indeed decrease the potential volume of the lake. We have mentioned this now in P5L9 / P4L6.

**P4, L2: Two surge periods is probably not sufficient to confirm 'a constant return period' especially over longer timescales, unless there are earlier indications of similar return period.**

The earlier surges – if we can take the main floods from upper Shimshal as a proxy - happened in 1979, 1960, 1944, 1923, 1901 or 1904 and possibly 1882 which corresponds to return periods of 22 (19), 19 (22), 21, 16, 19, 20 and 18 years from the end of the 19[th] century until today (Hewitt and Liu, 2010). We have added this reference to support the claim made.

**P4, L5: The observation of very different behaviour of the lower and upper parts of the tongue, separated by a steep part of bedrock at 12km, is interesting and has been observed on other surging glaciers (Quincey et al. 2014, Round et al. 2017). Possible questions to discuss here are whether there is something about the lower part of the tongue that leads it to experience such extreme changes in behaviour, or what the**

**significance of the steep section at 12km may be, or the significance of the avalanche mass deposits?**

There is no data available on avalanche accumulation and we believe it would likely not be enough mass to argue for a considerable influence on a surge. However we now follow the discussion described in (Quincey and Luckman, 2014; Round *et al.*, 2017), which emphasizes the role of local topography into surge behavior.

**P4, L12-13: Couldn't the increased pressure and 'tipping point' reached at the end of the quiescence also initiate the surge through collapse of the subglacial drainage system or failure of subglacial till? Is it possible to distinguish between these processes with the available data, or is there some other indication leading to the conclusion of a switch from cold to temperate basal conditions?**

We have referred to both mechanisms, but with the data available we are not able to separate them. However, we have made an estimate of deformation contributing to the observed velocities, which shows that that during the surge it must be primarily the basal motion that dominates flow, exemplified by the low values for ice deformation (P5L27 / P4L22ff). During quiescence in the lower part the switch from cold to temperate could have a more sizeable contribution.

**P4, L12-13: Do you mean this switch from cold to temperate based applies to the upper part of the tongue with the gradual acceleration, or lower part of the tongue with the sudden surge acceleration? Is it feasible that the velocities during the assumed cold based phase be purely due to ice deformation?**

We have considerably revised this section and in particular have pointed out that the thermal switch hypothesis likely pertains to the steeper section and below only.

**P4, L14: Quincey and Luckman (2014) suggested both the 'thermal switch' or 'subglacial drainage' as possible controls and didn't seem to have enough evidence to conclude one way or the other.**

The reviewer is right that their findings were pointing not to one or the other specifically. We rephrased this and avoided the suggestions that the thermal switch is the dominating driver.

**P4, L20: Did the velocity results show a parabolic velocity profile across the tongue during the quiescence? This wasn't mentioned in section 3 but would be interesting.**

Indeed a parabolic profile was observed during quiescence and this is now mentioned in the text at P5L40 / P4L30.

**P4, L21: The peak velocities of this surge are really incredibly high, as is the magnitude of the acceleration! A mention some of the feedback processes which could allow such extreme basal sliding velocities could be informative. Do you think subglacial till deformation plays much of a role?**

We have now added a rough quantification of ice deformation following (Round *et al.*, 2017), which shows that while it may episodically important, basal flow is likely to be the main driver. These feedback processes are indeed an interesting topic to be investigated in regard to this extreme acceleration, but so far we have no access to any kind of data (or modelling like (Damsgaard *et al.*, 2015) to support such claims. We have however added a suggestion for future investigation in this regard in P5L3 / P4L34.

**P4, L26: I'm not sure how the increased resolution and overpass frequency of the Planet satellite data have led to better understanding of the surge. Is it the ability to resolve the peak velocity over shorter time frame or observation of more temporal fluctuations or spatial patterns (e.g. transverse variation) in velocity? If so then this should be discussed somewhere.**

The main advantage of these satellite images is the high overpass frequency which improves the temporal resolution. This we have now emphasized by showing all velocity profiles for the surge itself in Figure 1 and have additionally discussed this advantage in Section 4.

**Figures 1 and 2: The right hand panels show the inferred glacier bed elevation, however it would make sense to also show the observed glacier surface elevation. Showing the surface elevation from the 2011 and 2017 DEMs would provide an additional visualization of the mass redistribution, and if shading or dashed lines are used the readability of the plot shouldn't be affected.**

We have made this change accordingly.

**Figures 1 and 2: The maps should be in some way georeferenced.**
We have tried to add a grid to the maps, but since we use a rotated north inclusion of the grid makes things unclear. We believe that the glacier coordinates we now provide in the text and the use of the RGI outline provide sufficient georeferencing for the reader.

**Figure 3: Very nice to have some photos from the ground, but maybe indicate the date (month)**

Thank you for the suggestion, we have added the date.

**Figure 3, L3: The traced 'centreline' would more appropriately be referred to as 'former centreline' or 'former medial moraine'**

Thank you for the suggestion, we have changed it accordingly.

**Figure 3, L8: I would say the tongue below the dashed green line "showed no change during the surge" rather than "remained stable".**

Thank you for the suggestion, we have changed it accordingly.

**Supplement Table S1: How many images were used from each Satellite? It would be interesting to have this information about the potential temporal resolution of the data.**

We have now provided this information in Table S1 in the Supplementary Material.

**Supplement Table S1 (DEM data): This table should be labelled Table S2.**

Thanks for pointing this out, we have changed it.

**Supplement Table S2: The SRTM from 2000 should also be shown here as it was also used for the surface elevation analysis.**

Thanks, we have added this.

[revised manuscript text omitted]

**1 Data**

**Table S1: LANDSAT/Planet Data for Velocity Datasets and Animation. All scenes  were acquired and visually pre-selected for cloud cover, snow cover and image quality over the glacier outline. The numbers in the first column refer to the actual pairing to derive velocities in Table S3.**

| COSI-Corr pair (Table S3) | Satellite / Scene | Band | Resolution | Acquisition Date |
|---|---|---|---|---|
| 1 | L7 LE07_L1TP_149035_20000911_20170210_01_T1 | 8 (Panchr.) | 15 m | 11/09/2000 |
| 1 / 2 / 3 / 4 | L7 LE07_L1TP_149035_20030531_20170125_01_T2 | 8 (Panchr.) | 15 m |  31/05/2003 |
| 2 / 6 | 2008_0925_20161029_01_T1LT05_L1TP_149035_ | 7 (SWIR) | 30 m | 25/09/2008 |
| 3 / 5 | 2010_1017_20161012_01_T1LT05_L1TP_149035_ | 7 (SWIR) | 30 m | 12/10/2010 |
| 4 | 2011_0918_20161006_01_T1LT05_L1TP_149035_ | 7 (SWIR) | 30 m | 18/09/2011 |
| 5 | 2009_0928_20161025_01_T1LT05_L1TP_149035_ | 7 (SWIR) | 30 m | 28/09/2009 |
| 6 / 7 | 2010_1017_20161012_01_T1LT05_L1TP_149035_ | 7 (SWIR) | 30 m | 17/10/2010 |
| 7 |  52011_0918_20161006_01_T1LT05_L1TP_149035_ | 7 (SWIR) | 30 m |  18/09/2011 |
| 8 | LC08_L1TP_149035_20130518_20170504_01_T1 | 8 (Panchr.) | 15 m | 18/05/2013 |
| 8 / 9 |  7LC08_L1TP_149035_20140910_20170419_01_T1 | 8 Panchr.) | 15 m |  10/09/2014 |
| 9 / 10 | LC08_L1TP_149035_20150828_20170405_01_T1 | 8 (Panchr.) | 15 m | 28/08/2015 |
| 10 / 11 | LC08_L1TP_149035_20160510_20170325_01_T1 | 8 (Panchr.) | 15 m | 10/05/2016 |

| | | | | |
|---|---|---|---|---|
| 11 / 12 | LC08_L1TP_149035_20161001_20170320_01_T1 | 8 (Panchr.) | 15 m | 01/10/2016 |
| 12 / 13 | LC08_L1TP_149035_20161220_20170315_01_T1 | 8 (Panchr.) | 15 m | 20/12/2016 |
| 413 / 16 |  8LC08_L1TP_149035_20170427_20170515_01_T1 | 8 ()Panchr.) | 15 m |  27/04/2017 |
| 514 | Planet Mosaic | Optical  | 3 m | 30/12/2016  |
| 14 / 15 | Planet Mosaic | Optical | 3 m | 16/04/2017 |
| 15 | Planet Mosaic | Optical | 3 m | 27/04/2017 |
| 16 / 18 | LC08_L1TP_149035_20170513_20170525_01_T1 | 8 (Panchr.) | 15 m | 13/05/2017 |
| 17 | Planet Mosaic | Optical | 3 m | 10/05/2017 |
| 17 / 19 | Planet Mosaic | Optical | 3 m | 25/05/2017 |
| 18 / 21 | LC08_L1TP_149035_20170529_20170615_01_T1 | 8 (Panchr.) | 15 m | 29/05/2017 |
| 19 / 20 | Planet Mosaic | Optical | 3 m | 29/05/2017 |
| 20 / 22 | Planet Mosaic | Optical | 3 m | 03/06/2017 |
| 21 / 30 | LC08_L1TP_149035_20170801_20170811_01_T1 | 8 (Panchr.) | 15 m | 01/08/2017 |
| 22 / 23 | Planet Mosaic | Optical | 3 m | 12/06/2017 |
| 23 / 24 | Planet Mosaic | Optical | 3 m | 19/06/2017 |
| 24 / 25 | Planet Mosaic | Optical | 3 m | 24/06/2017 |
| 25 / 26 | Planet Mosaic | Optical | 3 m | 27/06/2017 |
| 26 / 27 | Planet Mosaic | Optical | 3 m | 08/07/2017 |
| 27 / 28 | Planet Mosaic | Optical | 3 m | 21/07/2017 |
| 28 / 29 | Planet Mosaic | Optical | 3 m | 26/07/2017 |
| 29 / 31 | Planet Mosaic | Optical | 3 m | 01/08/2017 |
| 30 / 32 | LC08_L1TP_149035_20170817_20170825_01_T1 | 8 (Panchr.) | 15 m | 17/08/2017 |
| 31 / 33 | Planet Mosaic | Optical | 3 m | 19/08/2017 |
| 32 | LC08_L1TP_149035_20170918_20170929_01_T1 | 8 (Panchr.) | 15 m | 18/09/2017 |
| 33 | Planet Mosaic | Optical | 3 m | 12/09/2017 |

**Table S2: DEM Data**

| | Satellite / Product ID | Resolution | Acquisition Date | Reference |
|---|---|---|---|---|
| 1 | ASTER / AST_L1A.003:2253120815 | 30 m | 21/05/2017 | (NASA LP DAAC, 2017) |
| 2 | TanDEM-X / TDM1_DEM__04_N36E075_DEM | 12 m | Multiple during 2011 | DLR |
| 3 | SRTM | 1 arc second (~30 | 02/2000 | |

| | | | m) | | |
|---|---|---|---|---|---|

**Table S3: Velocity values [m a$^{-1}$] from all data products for the period between 2000 and 2017. Rows are km along the glacier tongue starting at the terminus. Columns are time steps and associated satellite products as described in Table S1. Color code corresponds to quiescence (green), build up (yellow) and surge (red).**

| Location | 1 L7 | L7/L5 | L7/L5 | L7/L5 | 5 L5 | 6 L5 | 7 L5 | 8 L8 | 9 L8 | 10 L8 | 11 L8 | 12 L8 |
|---|---|---|---|---|---|---|---|---|---|---|---|---|
| 1 | 4 | 2 | 2 | 2 | 3 | 3 | 3 | 2 | 1 | 3 | 2 | 19 |
| 2 | 5 | 2 | 3 | 2 | 2 | 2 | 3 | 2 | 1 | 3 | 3 | 17 |
| 3 | 8 | 2 | 2 | 2 | 2 | 4 | 6 | 2 | 2 | 5 | 4 | 19 |
| 4 | 12 | 2 | 2 | 2 | 3 | 6 | 9 | 9 | 6 | 7 | 6 | 16 |
| 5 | 6 | 2 | 2 | 1 | 1 | 4 | 7 | 6 | 4 | 7 | 9 | 10 |
| 6 | 5 | 2 | 3 | 2 | 1 | 3 | 6 | 5 | 10 | 16 | 35 | 82 |
| 7 | 9 | 2 | 2 | 1 | 2 | 5 | 9 | 19 | 28 | 41 | 129 | 286 |
| 8 | 7 | 3 | 3 | 2 | 3 | 9 | 12 | 42 | 53 | 68 | 216 | 401 |
| 9 | 6 | 3 | 4 | 4 | 5 | 12 | 15 | 54 | 62 | 77 | 250 | 430 |
| 10 | 11 | 2 | 27 | 11 | 14 | 16 | 18 | 83 | 89 | 92 | 297 | 435 |
| 11 | 13 | 2 | 11 | 21 | 14 | 11 | 29 | 96 | 96 | 102 | 296 | 417 |
| 12 | 13 | 1 | 3 | 18 | 17 | 6 | 25 | 96 | 105 | 107 | 298 | 384 |
| 13 | 28 | 1 | 2 | 12 | 6 | 6 | 15 | 120 | 125 | 121 | 314 | 359 |
| 14 | 41 | 1 | 4 | 11 | 6 | 8 | 9 | 140 | 139 | 133 | 305 | 330 |
| 15 | 52 | 1 | 13 | 12 | 5 | 5 | 6 | 147 | 152 | 148 | 306 | 297 |
| 16 | 61 | 2 | 60 | 38 | 5 | 2 | 4 | 162 | 167 | 162 | 308 | 272 |
| 17 | 70 | 2 | 29 | 51 | 4 | 5 | 9 | 164 | 185 | 166 | 332 | 254 |
| 18 | 79 | 2 | 73 | 43 | 2 | 5 | 6 | 76 | 197 | 206 | 331 | 232 |
| 19 | 111 | 2 | 65 | 75 | 21 | 7 | 6 | 198 | 204 | 199 | 296 | 240 |
| 20 | 111 | 1 | 26 | 33 | NA | 7 | 5 | 201 | 210 | 202 | 273 | 239 |
| 21 | 53 | 1 | 19 | 25 | NA | 12 | 5 | 144 | 224 | 213 | 275 | 240 |
| 22 | 57 | 1 | 2 | 10 | NA | 13 | 5 | 146 | 254 | 239 | 292 | 254 |
| 23 | 47 | 1 | 3 | 2 | NA | 6 | 4 | 193 | 246 | 232 | 279 | 255 |
| 24 | 22 | 1 | 3 | 2 | NA | 7 | 4 | 126 | 211 | 173 | 217 | 258 |
| 25 | 38 | 1 | 3 | 2 | NA | 9 | 3 | 41 | 193 | 197 | 274 | 292 |
| 26 | 51 | 1 | 3 | 2 | NA | 7 | 4 | 81 | 287 | 461 | 400 | 320 |

| Location | 13 L8 | 14 P | 15 P | 16 L8 | 17 P | 18 L8 | 19 P | 20 P | 21 L8 | 22 P | 23 P | 24 P | 25 P | 26 P | 27 P | 28 P | 29 P | 30 L8 | 31 P | 32 L8 | 33 P |
|---|---|---|---|---|---|---|---|---|---|---|---|---|---|---|---|---|---|---|---|---|---|
| 1 | 15 | 12 | NA | 12 | 39 | 45 | 166 | 55 | NA | NA | NA | NA | NA | 31 | 19 | 61 | 61 | NA | 40 | NA | 5 |
| 2 | 13 | 16 | NA | 10 | 54 | 28 | 185 | 80 | 5 | 880 | 495 | 1510 | 968 | 196 | 320 | 380 | 229 | 89 | 190 | 46 | 115 |
| 3 | 17 | 27 | 85 | 38 | 729 | 273 | 1523 | 1487 | 125 | 2452 | 1788 | 1768 | 2391 | 1036 | 1047 | 874 | 643 | 172 | 373 | 96 | 215 |
| 4 | 151 | 86 | 805 | 890 | 2892 | 1649 | 3192 | 2765 | 406 | 2502 | 2013 | 1879 | 2727 | 1510 | 1358 | 901 | 773 | 333 | 392 | 154 | 206 |
| 5 | 538 | 308 | 2486 | 2681 | 3701 | 2856 | 4000 | 3432 | 1188 | 2936 | 2454 | 2211 | 3089 | 1616 | 1451 | 992 | 936 | 381 | 405 | 147 | 189 |
| 6 | 527 | 304 | 2943 | 2719 | 3643 | 3145 | 4161 | 3591 | 1269 | 3124 | 2549 | 2392 | 3236 | 1723 | 1509 | 1003 | 943 | 357 | 355 | 111 | 140 |
| 7 | 1319 | 331 | 3732 | 4466 | 5034 | 4732 | 5242 | 4469 | 2145 | 3825 | 3138 | 3000 | 4018 | 2132 | 1902 | 1276 | 1200 | 409 | 421 | 120 | 153 |
| 8 | 1002 | 275 | 3715 | 4642 | 4836 | 4756 | 5167 | 4417 | 2392 | 4056 | 3284 | 2979 | 3972 | 2200 | 1943 | 1230 | 1203 | 351 | 361 | 84 | 120 |
| 9 | 1543 | 282 | 3581 | 4607 | 4837 | 5105 | 4930 | 4280 | 2149 | 3721 | 3138 | 2832 | 3903 | 2096 | 1864 | 1174 | 1113 | 286 | 304 | 54 | 81 |
| 10 | 1842 | 300 | 3390 | 4375 | 4435 | 4665 | 4883 | 4159 | 1766 | 3595 | 3095 | 2805 | 3943 | 2125 | 1811 | 1200 | 1139 | 281 | 285 | 51 | 68 |
| 11 | 684 | 273 | 3317 | 3980 | 4633 | 4809 | 5011 | 4288 | 2193 | 3759 | 3162 | 2887 | 4073 | 2172 | 1845 | 1235 | 1159 | 302 | 306 | 51 | 80 |
| 12 | 991 | 333 | 2860 | 3747 | 4366 | 4611 | 4526 | 3948 | 2317 | 3521 | 2937 | 2665 | 3830 | 2049 | 1703 | 1094 | 1135 | 285 | 274 | 44 | 80 |
| 13 | 1395 | 314 | 2924 | 2663 | 4018 | 4247 | 3682 | | 1774 | 3361 | 2792 | 2540 | 3604 | 1998 | 1628 | 1039 | 1105 | 281 | 281 | 48 | 77 |
| 14 | 947 | 318 | 2664 | 2696 | 3508 | 3350 | 3954 | 3506 | 1830 | 3157 | 2643 | 2604 | 3472 | 1952 | 1556 | 1001 | 1058 | 284 | 284 | 68 | 110 |
| 15 | 1067 | 310 | 2487 | 2426 | 3065 | 3400 | 3321 | 3147 | 1917 | 2819 | 2309 | 2220 | 3193 | 1753 | 1403 | 924 | 1006 | 288 | 299 | 76 | 129 |
| 16 | 1083 | 336 | 2860 | 2193 | NA | 3199 | 3257 | 3107 | 1772 | 2865 | 2370 | 2244 | 3160 | 1774 | 1473 | 948 | 1024 | 331 | 333 | 115 | 168 |
| 17 | 1029 | 386 | 2389 | 1816 | NA | 2684 | 2983 | 2844 | 1628 | 2659 | 2258 | 2092 | 2971 | 1682 | 1403 | 944 | 1012 | 378 | 403 | 194 | 247 |
| 18 | 611 | 346 | NA | 1320 | NA | 1998 | 2368 | 2281 | 1209 | 2142 | 1851 | 1817 | 2553 | 1419 | 1202 | 886 | 873 | 493 | 404 | NA | NA |
| 19 | 929 | 379 | NA | 1403 | NA | 1938 | 2059 | 2072 | 1447 | 1982 | 1738 | 1778 | 2452 | 1393 | 1196 | 923 | 891 | NA | 453 | NA | NA |
| 20 | 875 | 422 | NA | 1083 | NA | 1729 | NA | 1807 | 1523 | 1920 | 1852 | 1856 | 2459 | 1383 | 1171 | 887 | 933 | 492 | 480 | NA | NA |
| 21 | 571 | 374 | NA | 891 | 2227 | 1334 | NA | NA | 1324 | 1633 | 1753 | 1340 | 2085 | 1194 | 1001 | 751 | 852 | 363 | 431 | NA | NA |
| 22 | 565 | 328 | 871 | 784 | 1881 | 1076 | NA | NA | 1424 | 1453 | 1509 | 1247 | 1935 | 1101 | 940 | 718 | 817 | 348 | 462 | NA | NA |
| 23 | 413 | 305 | 657 | 626 | 1674 | 1029 | NA | NA | 1457 | 1173 | 1190 | 987 | 1487 | 928 | 781 | 616 | 715 | 330 | 411 | NA | NA |
| 24 | 279 | 321 | 654 | 532 | 1388 | 603 | NA | NA | 740 | 962 | 1019 | 813 | 1219 | 793 | 671 | 570 | 651 | 332 | 364 | NA | NA |
| 25 | 336 | 327 | 1004 | 378 | 947 | 315 | NA | NA | 823 | 905 | 756 | 1058 | 722 | 638 | 537 | 642 | NA | 355 | NA | NA | NA |
| 26 | 296 | 350 | 1155 | 296 | NA | 318 | NA | NA | NA | NA | NA | NA | NA | NA | NA | NA | NA | NA | NA | NA | NA |

**2 Model Settings and Uncertainties**

**2.1 Velocities**

COSI-Corr is sensitive to chosen initial window sizes as well as window steps (Leprince et al., 2007). In this study we work with different satellite products in respect to resolution and band quality – from the 30 m bands of the initial LANDSAT MSS Satellite to the 3 m optical product of Planet – which made different setups necessary. For the 30 m bands of Landsat MSS and Landsat-5 we used an initial window (W) of 128 pixels, a final window (F) of 16 pixels and a step size of 2 pixels (d) (W128-F16-d2). For Landsat- 7 and Landsat- 8 as well as Planet imagery we used a W128-F16-d4 setting while for surge events, when displacement is substantial or imagery is far apart in time, a W256-F16-d8 is used. We used the Non-Local Means Filter of COSI-Corr (Ayoub et al., 2009) to smooth the gridded data. Velocities measured on stable off-glacier terrain were used to assess the validity of the on-glacier data. The Landsat-MSS off-glacier velocities are in the same range as on-glacier velocities, which makes the COSI-Corr approach not suitable for this data. Off-glacier displacement based on Landsat- 5 data was between $2 - 5$ m a$^{-1}$, and this is sufficiently accurate to investigate the build-up and surge phase where velocities are generally one order of magnitude higher. Landsat- 7 and 8 as well as Planet data used in the analysis from 2013 onwards generally show off-glacier displacements of $2 - 3$ m a$^{-1}$ for imagery multiple days to weeks apart, which corresponds to the likely error identified by (Luckman et al., 2007)(Luckman et al., 2007). To make sure that noise, resulting from errors in the co-registration process, is not included in the data analysis, we discard all pixel values with a signal-to-noise ratio smaller than 0.75, following (Kraaijenbrink et al., 2016). As large displacements during a surge are picked up as noise by the algorithm in many cases, this constraint had to be loosened for surge peaks. In these cases patches on the surface that showed erratic behaviour (no uniform direction, large variability in velocities on a small area) were discarded visually.

**3 2.2 DEMs**

We compare the offset between the respective DEMs in relatively flat valley areas where all 3 DEM products are available (Figure S1a). Both TanDEM-X and the ASTER DEM are of less quality in steep terrain, however the 90$^{th}$ percentile of slope values on the investigated glacier surface is 10°, below which the quality is generally acceptable. We therefore exclude all values from the test areas with a slope above 10°. This results in a median offset between the TanDEM-X and the SRTM in the test areas of -24.8 m ($\sigma = 8.5$ m, Figure S1b), which is caused by the different geoids of the datasets, WGS84 and EGM96 respectively. Between the ASTER and the TanDEM-X an offset of -12.4 m ($\sigma = 2.3$ m) is found and used for correction on the ASTER (Figure S1c).

[Figure]

**Figure S1: (a) Flat stable areas chosen in the catchment used for comparison of the DEM. Blue shaded areas are glacier tongues. (b) Difference between TanDEM-X and adjusted SRTM. (c) Difference between corrected ASTER and TanDEM-X.**

**3 Lake Volume Calculation**

The volume of the lake was calculated by (1) deriving lake perimeters from the orthophotos, (2) draping them over the TanDEM-X digital elevation model, (3) taking the lake level as the 90[th] percentile of all elevation values (as the polygon of the perimeter does not have a continuous value due to the inaccuracies of the DEM), and (4) deriving the difference between this plane and the DEM (Figure S2). The lake volumes fit well with the exponential function derived by (Cook and Quincey, 2015) for lower volumes. As (Cook and Quincey, 2015) describe for ice-dammed lakes (Fig.4 therein), the curve steepens, i.e. volume increases faster than area, for larger areas, which is true for this lake as well. However for much greater extents (as observed in 2000) the relation does not hold anymore as areas increase faster due to the lake flooding a very shallow alluvial fan at the confluence of the Vijerab and the Khurdopin valley (see green and red markers in Figure S2b).

[Figure]

**Figure S2: (a) Lake areas derived from orthophotos and associated lake levels. (b) Relating lake area to lake volume. The black dots are values from 2017, the green marker is the lake in May 2000 and the red markers are projected areas and volumes with possible increase in lake level height 10 m above the value measured in 2000. The solid line is a relation for Volume-Area scaling found by (Cook and Quincey, 2015) which fits well for the observations in 2017, but overestimates for larger lake areas.**

**4 Supplementary Animation of all Landsat Scenes**

 Using all available Landsat imagery we compiled an animation over all scenes with suitable image quality (SupplementaryMaterial.zip). The images were not enhanced but comprise the raw rasters used for the  velocity analysis.

---

## Author Comment (AC2) · 11 Oct 2017

Response to Reviewer

Manuscript: Brief Communication: The Khurdopin glacier surge revisited – extreme flow velocities and formation of a dammed lake in 2017

**Reviewer: D.Quincey**

We greatly appreciate the concerns raised by the Reviewer and respond to each of them below. Original comments by the reviewer are in bold, followed by our response. Note that the page and line number are always given twice, once for the document with markups which is provided at the end of the Response, and once to the revised manuscript without markups, which will be provided later.

The key take-home message is currently a bit hidden. It seems to me that the new findings are: 1. that the surge return period appears to be of the order of 20 years (whilst acknowledging that n=2); 2. that surge velocities may be even faster than previously realised – implications for erosion and sediment transport; 3. that there may be a topographic control on this particular surge (but this needs much greater discussion – see following point); 4. that the ice-marginal lake is posing a hazard to local communities.

If the abstract and the conclusions could be modified to give the key message much greater prominence the manuscript would have greater impact. 2. The relevance of the steep bed topography at 12-km needs some further discussion/ explanation. Is the suggestion that it provides a control on surge dynamics? Or even that it is responsible for the spatial imbalance in flow? Presumably it doesn't provide a bottleneck to flow (I imagine the opposite if anything)? Is the modelled ice particularly thin above the step and potentially frozen? Some consideration of the possibilities would be a welcome addition. 3. There appear to be many more velocity datasets discussed in the text than presented in the figures. Is there a reason for not showing all of the velocity data? It would really help with visualising the evolution of the surge to have them all (or at least more than the current three) available. 4. The discussion of whether the surge is thermally or hydrologically triggered lacks real evidence so I would suggest toning it down or even removing it. It is likely that both thermal and hydrological processes will be at play as you infer in your own discussion. 5. There needs to be some uncertainty analysis of the dh/dt data. How well coregistered were the DEMs? Showing off-glacier areas of dh/dt data (and velocity data) would help here, as would the distribution of those values. This extra analysis would be a good addition to the Supplementary, with uncertainty shading added to the figures and an error range added to the values stated in the main text.

We would like to thank the reviewer for these suggestions and our response is found below.

We have adapted the abstract and especially the discussion and we have toned down the discussion on the switch hypothesis. We also extended this analysis of the steep topography section at km-12 and this reveals that ice deformation as well as hydrology may be important drivers. By now showing all velocity data of all time steps during the surge, we show the added value of these new satellite products, also for broader applications. We have also added a quantification of the DEM errors and we show this in figure S1 in the supplementary material.

Minor comments:

**P1 12: 'during a surge of the Khurdopin Glacier in 2017.' (also elsewhere, glacier should be Glacier where you are referring to it by name).**

Thanks for pointing this out. We have amended it throughout the manuscript.

**P1 15-16: I'm not sure there is evidence for a surge front in the data you show here?**

Thanks for pointing this out and as reviewer 2 has pointed out, the definition of front may have been used too subjectively here. Compared to other surges in the region and as you pointed out in (Quincey and Luckman, 2014), perhaps Khurdopin can be classified as not having a surge front. We have removed the reference to a front in the text.

**P1 19-20: do you show these surface observations? It's difficult for the reader to believe the extra lubrication suggestion without seeing evidence.**

In the Brief Communication format we are limited in terms of space and we have refrained from showing additional images. We agree however that this is an essential aspect and should be visualized. We have therefore included a 2-panel plot showing this distinct change in surface features in Fig 3.

**P1 26: this is maybe misleading: : : has an increase in frequency been reported? Or just an increase in number? And is that not because we have better and better data? Without repeat datasets (like those presented here) we can't say for sure whether frequency is increasing or not.**

We have based our statement mainly on the observation in these two papers, where especially (Copland *et al.*, 2011) argues that *"Given that our ability to detect surging using satellite imagery has remained essentially constant since the 1970s, we must therefore consider whether there have been changes in forcing over time."*. We agree however that this cannot be fully ascertained from the data we have and have therefore amended our text accordingly in P1L30 / P1L26. For Khurdopin specifically no increase in frequency can be found. If we can take the main floods from upper Shimshal as a proxy (1979, 1960, 1944, 1923, 1901 or 1904 and possibly 1882), this corresponds to return periods of 22 (19), 19 (22), 21, 16, 19, 20 and 18 years from the end of the 19th century until today (Hewitt and Liu, 2010).

**P1. 34-35: what do you mean by 'understanding regional glacier behaviour'? Is 'in order to advance knowledge of basal processes, non-steady flow more generally, and erosion and sediment transport in the region' perhaps a better justification?**

We agree that our explanation is too generic and we have replaced it including the suggestions made in P2L1 / P1L36.

**P2 3: name the glacier here, and also specify in the next sentence that it's the Khurdopin lake (not Kyagar) that has previously caused destruction.**

Thanks for the comment, we have adapted the text at P2L10 / P2L7.

**P2 8: maybe 'recent' is better than 'novel' here? Novel implies something a bit different about it.**

Thanks for pointing this out, we have changed this as suggested.

**P2 9: do you actually quantify the mass transfer somewhere? I don't see it.**

We have now provided an estimate of the volume flux based on the height changes in P4L16 / P3L26. As we do not have full glacier coverage with the DEMs (with the areas above the tongue being especially erroneous) and there is uncertainty in the DEMs, we cannot close the mass balance. It is therefore not possible to quantify the total volume/mass flux.

**P2 16: was the ASTER DEM derived by USGS? Or by the authors? In either case, some further information is required about its expected vertical accuracy and how well it performs against the TDX DEM.**

The ASTER DEM was generated through the AMES pipeline (Shean *et al.*, 2016) and we have now specified this in P2L27 / P2L19 and additionally added a discussion of vertical accuracy and our applied corrections in the Supplementary Material.

**P2 22: can you add the value (of mass loss) here?**

We have added a value of volume change in P4L16 / P3L26.

**P2 23: is it subglacially sourced for sure? I've always imagined it to be plucked from the spur where the two main tributaries meet.**

This may indeed also be the case. As Reviewer 1 suggested however this information is not strictly essential for this paper so we have removed the whole sentence.

**P2 26-27: is there a reason why you don't show these finer resolution velocity data?**

We have initially omitted showing these data for reasons of space in the Figures. However, also in accordance with Reviewer 1, and to further underline the value of the new Planet data we now show all velocity profiles during the surge in Figure 1 and provide all profile data in TableS3 in the Supplementary Material.

**P2 31: maybe reword to 'does not always allow the onset, peak and termination of the surge to be accurately identified, the data suggest that'?**

We have changed this as suggested.

**P2 32: not sure 'build-up' needs italics (here or at line 38)?**

This is indeed not necessary and we have amended it.

**P2 32-40 change to past tense here ('were below... and quickly rose...increased in1998... and peaked in spring 1999... phase lasted until... glacier had reached... was characterised by... velocities had reached...had further accelerated')**

Thanks for taking time for these comments, we have changed to past tense accordingly.

**P3 7: I'm not sure Figure 3 really supports this statement**

To support this statement we have now adapted Figure 3 to show the ponds before and crevasses after the surge.

**P3 9 and 14: if the lowermost 1 km of the glacier is not impacted by the surge is the length change not zero? What is meant by length change here (if not position of the terminus)?**

In Section 4 we refer to the changing part as "active tongue", which is the part of the tongue above the green line in Figure 3. Like on neighbouring Yazghil Glacier, Khurdopin has developed an ice-cored moraine at the terminus that downwastes but does not change position anymore since many years. As such it does not seem to be dynamically connected to the actual tongue anymore. We have clarified this in the text.

**P3 20-21: this is a long section between the commas – consider moving 'at rates comparable to those of the quiescent phase' before the first comma**

Also based on comments by Reviewer 1 we have revised and shortened this sentence.

**P3 32-38: it should be a short step to calculate the volumes from the DEM data – these values would be a valuable inclusion here.**

We have now added an estimate of ice volume gain in P4L16 / P3L26.

**P4 21: not quite true. The recent Hispar paper (doi:10.3390/rs9090888) by Paul et al. show comparable velocities**

Thanks for pointing this out, since this publication was published after our submission we have not included it. We have now referred to it accordingly in P6L5 / P4L37.

**P4 25-27: as far as I can tell the Planet imagery did not contribute to the data you present here other than the overview in Figure 3.**

Also following the main comment from the reviewer we have now made sure to emphasize the value of the Planet data for this study throughout this Section and in the Discussion. Additionally, we show all velocities derived from Planet pairs in figure 1 and in the Supplementary Material.

**Figure 1: some co-ordinates either here or in the text would help readers not familiar with the glacier to locate it.**

We now refer to figure 3 in figure 1 for a location and have provided coordinates in P2L4 / P2L1

**Figure 3: I'm not sure the wiggles are best described as 'centrelines'? Are they not the contorted medial moraines that have shifted position?**

Thanks for the suggestion, we have changed this.

**Supplementary: can you provide the image tile names in each case?**

We have now expanded table S1 in the Supplementary Material and provide details for each separate scene.

**Supplementary: Table S1 should be S2 in second case (and should SRTM be included here?).**

We have amended this.

**Supplementary: the animation is excellent. Should it not be referred to somewhere in the text (or it may go largely un-noticed)?**

We refer to this now in figure 1.

Jakob F. Steiner1, Philip D.A. Kraaijenbrink1, Sergiu G. Jiduc2, Walter W. Immerzeel1

1Utrecht University, Department of Physical Geography, PO Box 80115, Utrecht, The Netherlands 2Imperial College London, Centre for Environmental Policy, Faculty of Natural Sciences, SW7 1NA, London, United Kingdom

Correspondence to: Jakob F. Steiner (j.f.steiner@uu.nl)

Abstract. Glacier surges occur regularly in the Karakoram but the driving mechanisms, their frequency and its relation to a
 changing climate remain unclear. In this study, we use digital elevation models and Landsat imagery in combination with high-resolution imagery from the Planet satellite constellation to quantify surface elevation changes and flow velocities during a glacier surge in of the Khurdopin glacierGlacier in 2017. Results reveal that an accumulation of ice massyolume above a clearly defined steep section of the glacier tongue since the last surge in 1999 eventually leads to a rapid surge in May 2017 peaking with velocities above 5000 m a-1, which is among the fastest rates globally for a mountain glacier. The time series of Landsat imageryOur data reveals that velocities on the lower tongue increase steadily during a four-year build-

- up phase prior to the actual surge and that the surge front advances towards the terminus after theonly to then rapidly peak has passed on the upper tongue. The surgeand decrease again within a few months, which confirms earlier observations with a higher frequency of available velocity data. The surge return period between the reported surges remains relatively constant at 18 (1999 to 2017) and 20 (1979 to 1999) years respectively. It is hypothesized that the surge is mainly-initiated as
- 20 a result of increased pressure melting caused by ice accumulation, i.e. the thermal switch hypothesis. However, surface resulting in extremely high basal flow velocities, which may be further amplified by englacial hydrological processes. Surface observations show increased crevassing and disappearance of supra-glacial ponds, which could have led to increased lubrication of the glacier bed. Finally, we observe that the surging glacier blocks the river in the valley and causes a lake to form, which may grow in subsequent years and could pose threats to downstream settlements and infrastructure in
- 25 case of a sudden breach.

**1** Introduction**

Surging glaciers are not evenly distributed around the world's glaciated regions, but occur regularly under certain conditions (Sevestre and Benn, 2015). In the Karakoram, surges have been documented frequently since the end of the 19th century at numerous locations. In recent decades an increase in frequencyobserved surges has been reported (Copland et al., 2011;

- 30 Hewitt, 1969, 2007)-, however this has not been confirmed over larger areas and time periods and it could also be an artefact stemming from the increasing temporal availability of satellite imagery. Two general mechanisms driving surges are proposed: (a)-a build-up of ice massyolume during the quiescent phase in the reservoir zone of the glacier causing (a) increased basal shear stress resulting in till deformation at the glacier bed referred to as the *thermal switch hypothesis* (Clarke et al., 1984; Quincey et al., 2011)), and (b) a collapse of hydraulic channels causing a switch from efficient surface and
- 35 englacial drainage to sudden lubrication of the glacier bed referred to as the *hydrological switch hypothesis* (Kamb, 1987). Studies report surges in the region being controlled by both the first (Quincey et al., 2011) as well as the second mechanism (Mayer et al., 2011).

The Karakoram glaciers have received considerable scientific attention because of the anomalous regional mass balance (Kääb et al., 2015) and the large number of surging glaciers (Paul, 2015). Surging activity needs to be better understood in

order to furtheradvance our understandingknowledge of regional glacier behaviourice dynamic processes as well as glacially driven erosion and sediment transport in the region. Moreover, understanding of glacier surges is important as they may result in natural hazards that are due to the formation of ice dams and potential blockage of rivers.

Surges on Khurdopin glacierGlacier, located in the Shimshal valley in Northern Pakistan; (36°20'18"N, 75°28'3"E), have

- been documented to occur since the late 1800s and the most recent surges have occurred in 1979 and 1999 (Copland et al., 2011; Quincey et al., 2011; Quincey and Luckman, 2014; Rankl et al., 2014). These surges were characterized by a gradual increase of velocities before the peak of the surge (Quincey and Luckman, 2014). During the surge events, the lower tongue is pushed further into the valley and has blocked the Vijerab River on several occasions, resulting in an ice dammed lake, In the region, a similar to another process has been observed and well documented glacier in the regionfor Kyagar Glacier

[revised manuscript text omitted]
 50 to 80 m, consideringdepending on whether we assume the mean change between 2011 and 2017 and accounting fornet volume loss during the fact that elevation changes build-up phase between 2011 and 2016
- 15 were likely negative due to melt, at rates comparable to those of be zero or equal to the volume loss during the quiescent phase, between 2000 and 2011. Based on the elevation changes we find a net volume gain between 2011 and 2017 of  $520 \cdot 10^6 \text{ m}^3$  (+/-  $117 \cdot 10^6 \text{ m}^3$  based on the DEM accuracy) between the steep section and the part of the terminus where no more surface change is visible. Averaged over the entire glacier we estimate that the overall massyolume loss is slightly negative, (see surface elevation change in Figure 2), similar to what is reported by Bolch et al., (2017).

**20 5 Hydrology and Hazards**

5

10

The tongue of the Khurdopin glacier reaches across the main valley floor. As a consequence this glacier has blocked the local Vijerab river multiple times in recent decades, which has repeatedly caused considerable damages to settlements downstream (Iturrizaga, 2005). The blockage is caused as the tongue pushes towards the opposite headwall of the main valley (Figure 3). Most of the reported lake drainages were however not catastrophic and happened gradually as the river

- 25 water slowly erodes the glacier ice similar to other regional glacier lakes (Haemmig et al., 2014). From historic Landsat imagery it is obvious that a lake formed during the melt season in two consecutive years after the surge in 1999, likely because the added mass required considerable time to be eroded. In late April 2017, the lake formed at exactly the same location, growing quickly from 0.02 km2 at the beginning of May to 0.06 km2 one month later and more than 0.1 km2 on the 28th of June, reaching a lake depth of ca 2 m. Ice floes on the water surface indicate ice calving from the advancing tongue
- 30 and could pose an additional threat as they could block a drainage channel temporarily and create a sudden spill upon disintegration. Projected extents based on the DEM analysis correspond well to what was observed in 2000, when the lake was 0.7 km2 (Figure 3). Considering the height of the advanced glacier tongue between 15 m at the fringe and up to 80 m at the centre and a potentially large influx of sediments from Vijerab and Khurdopin subglacial drainage systems, we show potential lake extents that could reach up to 1 km2, possibly during the melt season of 2018 or 2019.
- 35 The tongue of the Khurdopin Glacier reaches across the main valley floor. As a consequence the glacier has blocked the local Vijerab river multiple times in the last century. The blockage is caused by the tongue that pushes towards the opposite headwall of the main valley (Figure 3). Most of the reported lake drainages were not catastrophic and they have rarely caused damages downstream beyond eroded fields and damaged bridges (Hewitt and Liu, 2010; Iturrizaga, 2005). From historic Landsat imagery it is obvious that a lake formed during the melt season in two consecutive years after the surge in
- 40 1999, likely because the added mass required considerable time to be eroded. In late April 2017, the lake formed at exactly the same location, growing quickly from 72000 m3 at the beginning of May to 1 · 106 m3 one month later and peaking at 2 · 106 m3 on the 28th 
[revised manuscript text omitted]

---

## Author Response (AR2)

Response to Editor

Manuscript: Brief Communication: The Khurdopin glacier surge revisited – extreme flow velocities and formation of a dammed lake in 2017

Utrecht, 21st of November 2017

Dear Editor,

Thank you very much for pointing out the issues below (in bold) and apologies for not covering all concerns entirely. We have followed your suggestions in detail and provide responses to all issues raised below. Note that the line numbering corresponds to the Manuscript Version without Markups.

With kind regards,

Jakob Steiner

Your response stresses at several instances that your submission is intended to be a Brief Communication (BC). With the reviewers' comments in mind I wonder if a BC is indeed the best format, or whether a normal paper would not make more sense. If you want to stick to a BC, then please make sure that the wording is very concise, unambiguous and fit and extensive elaborations for explanations could be omitted.

Considering that our paper is only focusing on a single surge event on one glacier we believe that the Brief Communication format is the most suitable. We describe a very recent event using a novel dataset and method (Planet Satellites, AMES pipeline for ASTER DEMs) with potential implications for subsequent studies and it is in line with previous work published as BC as well (Quincey and Luckman, 2014). This is in line with the description of what constitutes a Brief Communication at The Cryosphere. We have further condensed the text as detailed in the individual responses below and specifically at P1L18, P1L26, P3L4 and the Discussion. We pointed out the key findings more specifically in the Abstract (P1L18) and the Discussion (P4L19 – P5L1).

In the first two paragraphs rev. 2 provides a very structured criticism what should be improved in the manuscript. Your response falls short of showing what you indeed did to adequately consider them. E.g. point 2. steeper bedrock topography is briefly mentioned in the revision, but I do not find that this is the intended discussion the reviewer asked for. Please specify what you did change.

We have readdressed the comments of reviewer 2 and our response is provided pointwise below.

1. If the abstract and the conclusions could be modified to give the key message much greater prominence the manuscript would have greater impact.

We have adapted the abstract to point out the essential findings. These include the actual values found for the surge in 2017 (peak velocity, P1L14), rapid development of the surge (P1L15), confirmation of the return period (P1L17) and the potential of Planet (and ASTER) data to document and investigate surges because of a more frequent overpass than Landsat (P1L18ff; new edit in this version). We have removed the part on the triggers for the surge as they cannot be fully ascertained from the data but they are discussed now only in the Discussion (P4L19 – P5L1, new edits in this version).

2. The relevance of the steep bed topography at 12-km needs some further discussion/ explanation. Is the suggestion that it provides a control on surge dynamics? Or even that it is responsible for the spatial imbalance in flow? Presumably it doesn't provide a bottle-neck to flow (I imagine the opposite if anything)? Is the modelled ice particularly thin above the step and potentially frozen? Some consideration of the possibilities would be a welcome addition.

The coincidence of the steep bedrock and the turnover point for mass as well as the point of speedup of velocities is striking and warrants attention. We have no further (field) data to ascertain it drives the surge but we now provide an additional calculation including the slope and ice thickness that shows that it could indeed play a role in accelerating flow during quiescence and the early build up phase, resulting in a transition from a cold to a temperate thermal regime at the bed (P4L19ff). As the reviewer points out it is not a bottle neck and the ice thickness is in the same order of magnitude as elsewhere on the tongue. The relatively high slope is an indicator that ice deformation

plays a significant role during the early phase, while a switch from the frozen state to a temperate state must have occurred to explain the much higher velocities soon after. The flat velocity profiles during the actual surge phase then point to basal sliding being the dominant process. We now describe this in the lines following P4L19.

3. There appear to be many more velocity datasets discussed in the text than presented in the figures. Is there a reason for not showing all of the velocity data? It would really help with visualising the evolution of the surge to have them all (or at least more than the current three) available.

We have considered this by including all surge velocities available in Figure 1 and providing additional data in the Supplementary Material (Table S2).

4. The discussion of whether the surge is thermally or hydrologically triggered lacks real evidence so I would suggest toning it down or even removing it. It is likely that both thermal and hydrological processes will be at play as you infer in your own discussion.

See our response to the following comment by the Editor.

5. There needs to be some uncertainty analysis of the dh/dt data. How well coregistered were the DEMs? Showing off-glacier areas of dh/dt data (and velocity data) would help here, as would the distribution of those values. This extra analysis would be a good addition to the Supplementary, with uncertainty shading added to the figures and an error range added to the values stated in the main text.

We included a description of the DEM uncertainty based on an analysis of off-glacier areas in the supplemental material (section 2.2) and here we also described the corrections applied on the DEMs.

Regarding point 4., you toned down the role of the type of triggering mechanism, but it is still stressed in the abstract that you " ... hypothesize that the surge is initiated as a result of increased pressure melting caused by ice accumulation resulting in extremely high basal flow velocities."

I find such a statement unfit for an abstract. In fact, although it is in principle alright to have some hypothesis at the end of a paper, which cannot be fully falsified or verified (as long as that is clearly stated), in the present case I find that these two possible trigger mechanisms are insufficiently discussed in the MS.

My suggestions here are

a) say in the abstract what you did and what you did (or did not) find (evidence) and b) explain in more detail why you come to a certain conclusion, e.g. the statement on p4/l26 requires more explanation.

"... thermal switch hypothesis being at play, because ...".

Thank you for these suggestions. We have removed the hypothesis from the abstract and we have improved this part in the discussion by stressing the importance of the steep gradient as a possible focal point for the surge initiation (from P4L19). Using estimates for ice deformation we show that the bed is likely cold based during quiescence but switches to temperate in the steep section, which could explain the local initiation of the surge (Figure

1). The surge itself, with its extreme velocities and plug flow, is clearly characterized by basal sliding, further confirmed by large amounts of basal till emerging from the snout.

**Figure 1: Velocities calculated based on ice deformation** (Greve and Blatter, 2009; Round *et al.*, 2017) **in the top panel as well as observed velocities from the surface in the bottom panel, both along the full length of the tongue.**

**Your response to rev. 2 comment on p4., I26: "we...discussed this advantage in Section 4.": I did not see where you added or extended such a discussion, please clarify.**

This reference was misplaced and should have referred to Section 6 (P5L3), where we emphasize the value of the Planet data. We also emphasize this by now showing all velocities from the surge in Figure 1, which enable us to make statements about the location of the surge peak and rapid acceleration and deceleration (P2L41 and following).

**Figure 1:**

**a-c: Provide georeferenced coordinates (mandatory!). You don't need a grid, it would be sufficient to show a couple of lat/lon ticks/labels on the border of the figure.**

We have now added a georeferenced grid to the Figure.

**a-c: Add units of velocity**

We have added the units.

**cross correlating -> cross-correlating Landsat ...**

We have changed this.

Explain graph d please further in the caption, not too clear yet, especially the difference between the black lines and the red/yellow lines; e.g. by: "values for the quiescence have been multiplied by ..., values for the build-up ...". This leads to ambiguities. A figure should be understandable from the caption alone.

We have amended this in the caption to clarify this visualization. insert after (yellow): both, for Landsat and Planet.

We have added this.

Figure 2:

"Note that ... is much larger than ..."

In addition to showing the original series, why didn't you take this into account by extrapolating the series from 2000-2011 into 2016 and thus calculating more realistic changes in surface elevation? I appreciate that you want to stick simply to the dates of elevation observations, but the physical point you try to make could be emphasized in this way much more, e.g. the impact on erosion rates etc.

We have followed this suggestion and now show ice changes for the quiescent phase (2000 until 28th of August 2015) as one (blue) line and for the surge phase (between 28th of August of 2015 and 21 May 2017 (acquisition time of the ASTER DEM, the actual build up and surge) as a separate line (red). As a consequence the values in Figure 2 have changed substantially. Additionally we adapted the text corresponding to the Figure in P3L15. As a consequence the values for mass changes have changed in P3L24 according to the new time period and as they were calculated inaccurately in the previous manuscript.

**Figure S1a: add georeferenced coordinates.**

We have now added a georeferenced grid to the Figure.

Minor comments:

**p1/l32: remove extra )**

We have removed it.

**p2/l19: you use two numbers, 30 and 3 m, but do not explain what they mean. Resolution, pixel size, etc.**

We have added 'resolution' accordingly.

**p2/l37: remove extra )**

We removed this.

**p3/l9: Figures should be sorted according to order of appearance in the text.**

We have moved the section on crevassing and strain rates to the conclusion (P4L40) and we have removed a duplicate comment on the stable terminus. Now the order of the figures is restored.

**p3/l18: terminus -> apparent terminus**

We have added this.

pe/l19: add after moraine: ... dynamically decoupled from the active part of the glacier. That would take care of the rev.'s criticism.

Thanks for this suggestion. We have added it accordingly.

**p3/23,24: remove () from Gardelle citation.**

We have removed it.

p4/3: "Projected extents": explicitly clarifiy of what - the lake. of what? -> the lake. Unclear what you want to say here: if a lake would form again with a certain water level, then would you get this extent?

We agree this is not very clear we have therefore removed it.

Non-public comments to the Author: Please clearly respond to all of the above points

[revised manuscript text omitted]

---

## Author Response (AR3)

Response to Editor

Manuscript: Brief Communication: The Khurdopin glacier surge revisited – extreme flow velocities and formation of a dammed lake in 2017

Utrecht, 26th of November 2017

Dear Editor,

Thank you very much for taking the time to meticulously address all remaining issues. I have complied with all the requested changes.

With kind regards,

Jakob Steiner